# Limited Sensitivity of Circulating Tumor DNA Detection by Droplet Digital PCR in Non-Metastatic Operable Gastric Cancer Patients

**DOI:** 10.3390/cancers11030396

**Published:** 2019-03-21

**Authors:** Luc Cabel, Charles Decraene, Ivan Bieche, Jean-Yves Pierga, Mostefa Bennamoun, David Fuks, Jean-Marc Ferraz, Marine Lefevre, Sylvain Baulande, Virginie Bernard, Sophie Vacher, Pascale Mariani, Charlotte Proudhon, Francois-Clement Bidard, Christophe Louvet

**Affiliations:** 1Department of Medical Oncology, Institut Curie, PSL Research University, 75005 Paris and 92210 Saint Cloud, France; luc.cabel@curie.fr (L.C.); jean-yves.pierga@curie.fr (J.-Y.P.); 2Versailles-Saint-Quentin University, Paris-Saclay University, 92210 Saint Cloud, France; 3Circulating Tumor Biomarkers Laboratory, Institut Curie, PSL Research University, INSERM CIC 1428, 75005 Paris, France; Charles.Decraene@curie.fr (C.D.); charlotte.proudhon@curie.fr (C.P.); 4CNRS UMR144, Institut Curie, PSL Research University, 75005 Paris, France; 5Pharmacogenomics Unit, Department of Biopathology, Institut Curie, PSL Research University, 75005 Paris, France; ivan.bieche@curie.fr (I.B.); sophie.vacher@curie.fr (S.V.); 6INSERM U1016, Faculty of Pharmaceutical and Biological Sciences, Paris Descartes University, 75005 Paris, France; 7Paris Descartes University, 75005 Paris, France; 8Department of Medical Oncology, Institut Mutualiste Montsouris, 75014 Paris, France; mostefa.bennamoun@imm.fr; 9Department of GI Surgery, Institut Mutualiste Montsouris, 75014 Paris, France; david.fuks@imm.fr (D.F.); jean-marc.ferraz@imm.fr (J.-M.F.); 10Department of Pathology, Institut Mutualiste Montsouris, 75014 Paris, France; marine.lefevre@imm.fr; 11ICGex Next-Generation Sequencing platform, Institut Curie, PSL Research University, 75005 Paris, France; sylvain.baulande@curie.fr (S.B.); Virginie.Bernard@curie.fr (V.B.); 12Department of Surgery, Institut Curie, PSL Research University, 75005 Paris, France; pascale.mariani@curie.fr

**Keywords:** circulating tumor DNA, gastric cancer, monitoring, minimal residual disease

## Abstract

This study was designed to monitor circulating tumor DNA (ctDNA) levels during perioperative chemotherapy in patients with non-metastatic gastric adenocarcinoma. Plasma samples were prospectively collected in patients undergoing perioperative chemotherapy for non-metastatic gastric adenocarcinoma (excluding T1N0) prior to the initiation of perioperative chemotherapy, before and after surgery (NCT02220556). In each patient, mutations retrieved by targeted next-generation sequencing (NGS) on tumor samples were then tracked in circulating cell-free DNA from 4 mL of plasma by droplet digital PCR. Thirty-two patients with a diagnosis of non-metastatic gastric adenocarcinoma were included. A trackable mutation was identified in the tumor in 20 patients, seven of whom experienced relapse during follow-up. ctDNA was detectable in four patients (*N* = 4/19, sensitivity: 21%; 95% confidence interval CI = 8.5–43%, no baseline plasma sample was available for one patient), with a median allelic frequency (MAF) of 1.6% (range: 0.8–2.3%). No patient with available plasma samples (*N* = 0/18) had detectable ctDNA levels before surgery. After surgery, one of the 13 patients with available plasma samples had a detectable ctDNA level with a low allelic frequency (0.7%); this patient experienced a very short-term distant relapse only 3 months after surgery. No ctDNA was detected after surgery in the other four patients with available plasma samples who experienced a later relapse (median = 14.4, range: 9.3–26 months). ctDNA monitoring during preoperative chemotherapy and after surgery does not appear to be a useful tool in clinical practice for non-metastatic gastric cancer to predict the efficacy of chemotherapy and subsequent relapse, essentially due to the poor sensitivity of ctDNA detection.

## 1. Introduction

Gastric cancer displays significant global variation in incidence, with the highest rates observed in Eastern Asia, Eastern Europe and South America [1]. The use of preoperative and postoperative combination chemotherapy, in addition to surgery, has been shown to improve the overall survival of non-metastatic gastric cancer patients [2]. Current treatment guidelines recommend that chemotherapy should be administered for 2–4 months prior to surgery, and resumed thereafter, for a total of about 6 months of chemotherapy [3,4]. Despite this combined treatment, the long-term survival of non-metastatic gastric cancer remains limited, with a 36% 5-year overall survival rate [2]. While the pathological response to preoperative chemotherapy has been associated with relapse-free and overall survival [5], no biological or imaging tool is able to detect and quantify minimal residual disease, i.e., residual cancer cells after chemotherapy and/or surgery that are responsible for subsequent metastatic relapse.

Circulating tumor biomarkers have demonstrated their clinical validity in several non-metastatic cancer types either as baseline prognostic biomarkers, as a monitoring tool during preoperative chemotherapy and/or as a way to detect minimal residual disease [6,7,8,9,10,11,12,13,14,15,16,17,18,19,20].

This proof-of-concept study was designed to investigate the sensitivity and prognostic impact of circulating tumor DNA (ctDNA) detection in non-metastatic gastric cancer patients undergoing perioperative chemotherapy.

## 2. Results

### 2.1. Patients and Samples

Thirty-two patients, with a diagnosis of non-metastatic gastric cancer, were included in this study between June 2014 and October 2016. Seventeen (53%) of these patients had locoregional nodal dissemination at baseline imaging; other patient characteristics are shown in Table 1. Median follow-up was 26 months (range: 11–35 months). A trackable mutation was found in 20 tumor samples (63% of patients), most commonly *TP53* mutations (Table 2). All patients received preoperative FOLFOX-based chemotherapy, two patients also received trastuzumab and three patients received FOLFOX plus nab-paclitaxel. Seven patients obtained a pathological complete response (22%), including four of the 20 (20%) patients with a trackable mutation. Seven patients in the cohort with a trackable mutation experienced metastatic relapse during follow-up.

### 2.2. ctDNA Detection and Correlation with Patient Characteristics in Patients with a Trackable Mutation

Nineteen of the 20 patients with a trackable mutation were assessable for baseline ctDNA detection before preoperative chemotherapy (Figure 1). A blood sample was collected before surgery for 18 out of 19 patients and after surgery for 13 out of 19 patients (Figure 1, Table 2).

At baseline, four out of 19 patients had detectable ctDNA (>0.1%); the droplet digital PCR (ddPCR) technique displayed a sensitivity of 21% (95% CI = 8.5–43%) in this non-metastatic setting (Table 2). ctDNA detection was not significantly associated with baseline clinical characteristics—two patients had N+ disease (50% in ctDNA-positive versus 47% in ctDNA-negative, *p* = 1), none had HER2-positive disease (0% versus 11%, *p* = 1). Among the four patients with detectable ctDNA, median mutant allele frequency (MAF) was relatively low (1.6%, range: 0.8–2.3%). No correlation was observed between baseline ctDNA detection and relapse, as only one patient in the baseline ctDNA-positive group experienced relapse (*n* = 1/4, 25%) versus 6/15 patients in the ctDNA-negative group (40%), *p* = 0.52 (Fisher’s exact test).

After preoperative chemotherapy (i.e., before surgery), ctDNA was undetectable in all patients (0%, *n* = 0/18, 95% CI = 0%–18%), including those with detectable ctDNA at baseline (Figure 2). After surgery, ctDNA was detectable in one patient (*n* = 1/13, 95% CI = 1%–33%) with an MAF of 0.7%. No ctDNA was detected in this patient at diagnosis (before preoperative chemotherapy) or before surgery; he obtained a minimal pathological response (ypT3N3) and experienced early metastatic relapse 3 months after surgery. No ctDNA was detected after surgery in the available plasma samples from the other four patients who experienced a later relapse (at 9.3, 10.3, 18.5 and 26 months).

## 3. Discussion

In this study, we report that ctDNA can detected by customized ddPCR assays in 20% of non-metastatic gastric cancer patients prior to the initiation of preoperative chemotherapy.

The ctDNA detection rate for gastric cancers has been reported to be around 30–50% in non-metastatic gastric cancer patients, and the ctDNA detection rate has been correlated with the stage of the disease [11,22,23,24]. Only one study analyzed the monitoring of ctDNA in gastric cancers (other than the detection of methylation in cell-free circulating DNA (cfcDNA)), and showed that three out of 10 patients (30%) with TP53 mutations in primary tumors showed detectable TP53 mutation levels in the preoperative setting [24]. In our study, the 20% detection rate of ctDNA before therapy appears to be lower than that reported in other non-metastatic cancer types, despite using the same approach and ddPCR technique as those reported in other studies from our laboratory [13,17,20,25]. The lower detection rate in the present study could be explained by the systematic laparoscopic assessment of peritoneal metastasis, combined with extensive imaging workup to detect metastasis. This may increase the proportion of truly non-metastatic patients, as ctDNA detection is correlated with tumor burden [11,22,23]. More sensitive detection techniques, such as ultra-deep next-generation sequencing using a unique molecular identifier bar-coding strategy, may need to be used [26]. However, the mechanisms underlying this low detection rate have yet to be determined. By using a broader panel of genes for sequencing, such as whole-exome sequencing, more patients could have benefited from ctDNA monitoring (12 patients in this study could not be monitored). In addition, several trackable mutations could have been detected for each patient, which could increase the detection sensitivity. It is nevertheless important to monitor mutations considered as drivers, because passenger mutation evolution is less representative of tumor evolution. Also, patients with different tumor mutations may have a different clinical course, as these mutations can be prognostic. Another approach to ctDNA detection based on the detection of circulating tumor methylation could be of interest in this context [27,28]. In addition to ctDNA, other circulating tumor biomarkers such as circulating tumor cells (CTC) and exosomes might be also relevant. However, the detection rate of CTC, as reported in multiple studies [29,30,31,32], appears low in metastatic gastric cancer patients (e.g., less than half of 106 patients with advanced gastric cancer displayed ≥2 CTCs using CellSearch [33]), and precludes any clinical use in non-metastatic gastric cancer patients. Lastly, exosome detection has gained increasing interest, being potentially more sensitive than ctDNA in lung cancer [34], but only a few preliminary reports are available on gastric cancers [35,36].

The most striking results of this study are that ctDNA levels dropped markedly in most patients during chemotherapy. This is the first study to document this fall in ctDNA levels in gastric cancer patients, although similar ctDNA kinetics have been reported in other cancer types, such as breast [17] and rectal cancers [37]. This study highlights that ctDNA detection is not a reliable biomarker to predict relapse in the neoadjuvant/preoperative setting, despite the use of a highly sensitive technique (ddPCR), as ctDNA levels generally drop dramatically during chemotherapy, even in patients who subsequently experienced relapse. However, as demonstrated in triple-negative breast cancer in the neoadjuvant setting, the kinetics of non-detection of ctDNA could constitute a prognostic marker, but requires early sampling that was not performed in this study [17].

ctDNA detection after surgery has been shown to be a very useful biomarker to predict relapse in colon cancer [14] or melanoma [19]. All of these studies [14,15,16] also highlighted that the serial monitoring of ctDNA increased the sensitivity for prediction of relapse, compared to a single assay after surgery. However, most of these reports had a short follow-up [14,15,16,38] and relapse in the ctDNA-positive group mostly occurred during the first 12 months after surgery, as observed in one patient in this study. This limited time interval between ctDNA detection and onset of clinical relapse suggests that patients with detectable ctDNA already harbor micrometastatic disease, while the presence of limited minimal residual disease and/or non-proliferating disseminated tumor cells cannot be detected by means of the current technique. The question of whether residual ctDNA levels detected after therapy reflect the presence of metastases that are already growing (but not initially detected by imaging) or minimal residual disease (which may be quiescent) remains unresolved.

The main limitations of our proof-of-concept study are the small number of patients analyzed and the low rate of trackable mutations detected in the cohort due to the NGS panel available at our institution.

## 4. Material and Methods

### 4.1. Patients and Treatments

This prospective study (NCT02220556) included plasma samples from patients with non-metastatic gastric cancer treated at Institut Mutualiste Montsouris and Institut Curie (Paris, France). Eligibility criteria were: Histologically-proven gastric cancer; planned perioperative chemotherapy; absence of distant metastasis (CT scan, PET/CT and laparoscopy); no history of invasive cancer. Written informed consent to participate was obtained from all patients. Preoperative chemotherapy consisted of six cycles of FOLFOX (5-fluorouracil and oxaliplatin). In addition to FOLFOX, some patients received nab-paclitaxel in the context of a clinical trial, as well as trastuzumab in HER2-positive cancers. Surgical resection was performed after preoperative chemotherapy and consisted of complete or partial gastrectomy (or Lewis–Santy procedure whenever required). The same chemotherapy regimen was resumed after surgery whenever possible. Patient characteristics, treatment and outcomes were prospectively recorded. Tumor staging was performed according to the 7th UICC TNM classification [21] using endoscopic ultrasound, CT scan, PET/CT and laparoscopic staging. Follow-up included clinical evaluation and chest/abdomen/pelvis CT scan every 6 months during the first 3 years of follow-up, and annually thereafter.

### 4.2. Next-Generation Sequencing on Tumor Sample

Targeted next-generation sequencing was performed based on biopsy or gastrectomy, with a panel of 39 cancer-related genes (see Appendix A) using an Illumina HiSeq 2500 system. Library preparation was performed as reported previously [25]. A depth of coverage of >50 reads was required for variant calling, with thresholds of 1% for the alternate allele for the calling of SNVs/mutations, and 5% for indels. Raw reads were aligned on the reference human genome hg19 using the TMAP aligner (v0.3.7 Life Technologies). The variants were annotated using ANNOVAR and the following databases: COSMIC68, dbSNP137, 1000 genomes, ESP6500 and RefGene annotations. Only non-synonymous variants not observed in >0.1% of the population (1000 genomes and ESP6500) were identified as possible trackable somatic mutations. A trackable mutation in plasma was defined as a pathogenic variant in a driver gene, with an MAF greater than 1% in the tumor and coverage of at least 300×.

### 4.3. ctDNA Detection

In this study, blood samples were collected before the initiation of preoperative chemotherapy, before surgery and after surgery (<1 month). At each time point, 21 mL of blood was drawn in EDTA tubes (three tubes) and processed within 1 h to obtain 6–8 mL of plasma after centrifuging blood at 820× *g* for 10 min. Plasma was transferred to 2 mL tubes and centrifuged at 16,000× *g* for 10 min to remove debris, then stored at −80 °C until needed. Cell-free circulating DNA (cfcDNA) extraction was performed on 4 mL of plasma using the QiaSymphony SP instrument and the QIAsymphony Circulating DNA kit (Qiagen), an automated technique, according to the manufacturer’s protocol. cfcDNA was eluted into 60 µL of elution buffer and stored at −20 °C. cfcDNA was then subjected to ctDNA detection by ddPCR (droplet digital PCR), using the cfcDNA equivalent to 2 mL of plasma. Briefly, ddPCR assays matching the trackable somatic mutations found in tumor tissue were purchased from Bio-Rad () and the analysis was performed using the dedicated platform develop by Bio-Rad according to the manufacturer’s protocol. Briefly, for each sample, a total volume of 20 μL PCR reaction mixture was prepared with 10 μL 2× Supermix for Probes without dUTP (Bio-Rad), 1 μL 20× target primers/probe (Bio-Rad), 1 μL 20× wild-type primers/probe (Bio-Rad) and DNA sample/water qsp 20 µL. The PCR reaction mixture was portioned into droplets using the QX-100 Droplet Generator (Bio-Rad) according to the manufacturer’s instruction. The droplets were then transferred to a 96-well PCR plate and ddPCR was conducted using a C1000 Thermal Cycler (Bio-Rad) as follows: 95 °C for 10 min, 40 cycles of 94 °C for 30 s, 55 °C for 60 s and 10 min at 98 °C. The samples were then transferred to a Bio-Rad QX-100 droplet reader and analyzed based on fluorescence intensity. The mutant allele frequency (MAF) was calculated for each sample using QuantaSoft v1.7.4 software (Bio-Rad). Each test contained at least one negative control well with no DNA and the corresponding tumor. The performance and detection threshold of the ddPCR assays used in this study have been either reported elsewhere [17,25] or validated by the manufacturer (Bio-Rad).

All ddPCR probes were therefore tested on primary tumor biopsy, and all trackable somatic mutations identified by NGS were detectable with >1% MAF in all primary tumors with ddPCR (Table 2).

### 4.4. Statistical Analyses

This hypothesis-generating study had no prespecified power. For nonparametric analysis, chi-square or Fisher’s exact test were used for categorical variables, using GraphPad Prism.

### 4.5. Compliance with Ethical Standards

All applicable international, national, and/or institutional guidelines for the care and use of animals were followed. All procedures performed in studies involving human participants were in accordance with the ethical standards of the institutional and national research committee (ethical approval: Comité protection des personnes, CPP, approval 04/01/2018 N°29-15, study approval: Agence Nationale de Sécurité du Médicament et des produits de santé (ANSM) approval 03/04/2015 N°150221B-12) and with the 1964 Helsinki declaration and its later amendments or comparable ethical standards. Informed consent was obtained from all individual participants included in the study.

## 5. Conclusions

In conclusion, ctDNA monitoring during preoperative chemotherapy and after surgery does not appear to be a useful tool in clinical practice for non-metastatic gastric cancer to predict the efficacy of chemotherapy and subsequent relapse, essentially due to the poor sensitivity of ctDNA detection, despite using a highly sensitive method of ctDNA detection (MAF > 0.1%).

## Figures and Tables

**Figure 1 cancers-11-00396-f001:**
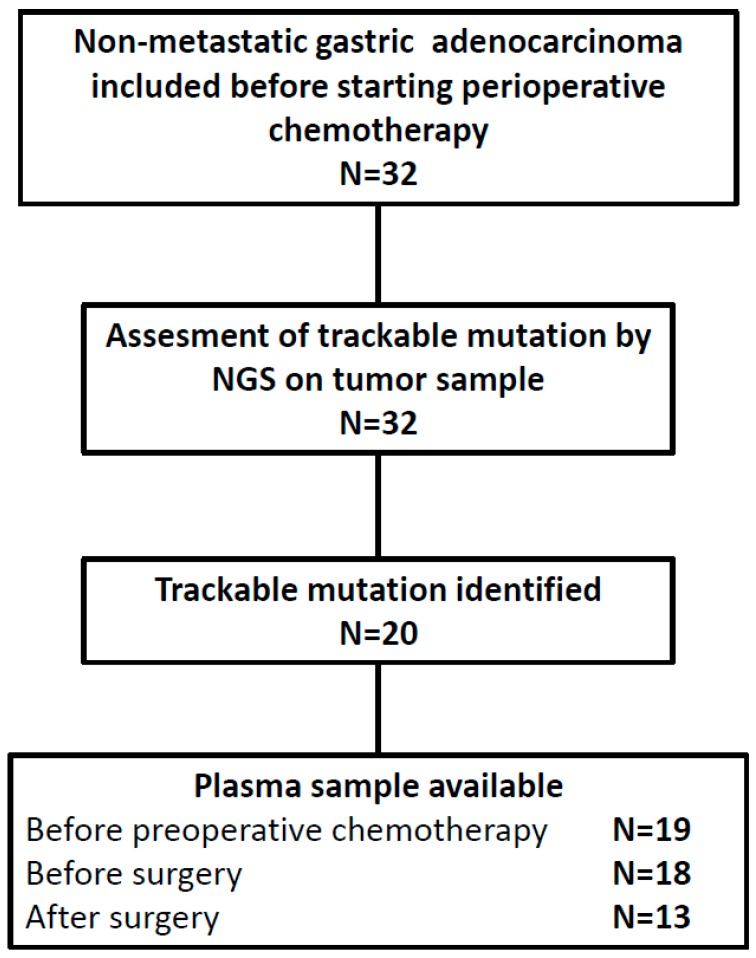
Study flowchart. NGS: next-generation sequencing.

**Figure 2 cancers-11-00396-f002:**
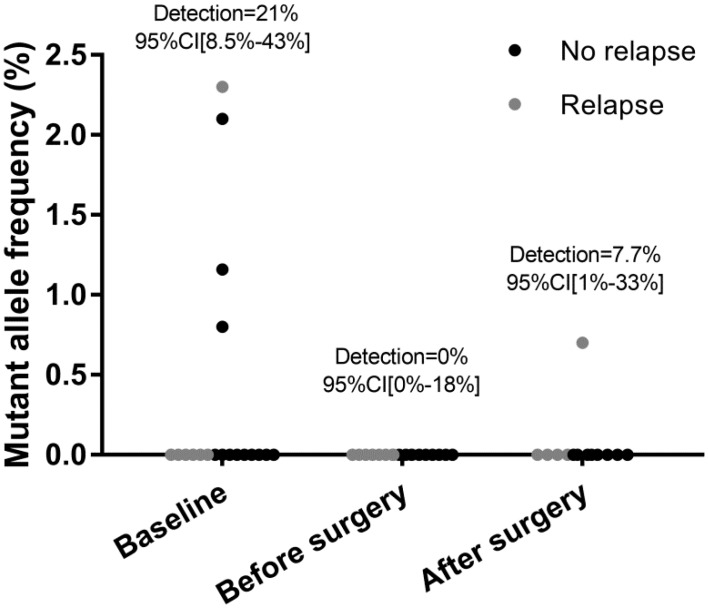
Circulating tumor DNA detection according to time point (baseline, before surgery, after surgery) by droplet digital PCR.

**Table 1 cancers-11-00396-t001:** Patient characteristics.

Characteristics	Whole Cohort *N* = 32	Cohort with a Trackable Mutation *N* = 19	Patients with Detectable Circulating Tumor DNA (ctDNA) at Baseline *N* = 4	Detectable versus Undetectable ctDNA at Baseline *p*-Value
**Age (years)**				
Median (range)	65 (36–78)	62 (45–78)	60 (45–70)	N.S.
**Gender**				
Male	24 (75%)	16 (84%)	4 (100%)	1
Female	8 (25%)	3 (16%)	0 (0%)	
**T stage (baseline)**				
T1	3 (9%)	1 (5%)	0 (0%)	0.77
T2	8 (26%)	4 (21%)	1 (25%)	
T3	18 (56%)	12 (63%)	2 (50%)	
T4	3 (9%)	2 (11%)	1 (25%)	
**N stage (baseline)**				
N0	15 (47%)	9 (47%)	2 (50%)	1
N+	17 (53%)	10 (53%)	2 (50%)	
**Histology**				
Intestinal	26 (81%)	17 (89%)	4 (100%)	1
Diffuse	6 (19%)	2 (11%)	0 (0%)	
**HER2 status**				
Positive	4 (13%)	2 (11%)	0 (0%)	1
Negative	28 (87%)	17 (89%)	4 (100%)	
**Localization**				
Cardia	20 (63%)	14 (74%)	4 (100%)	0.54 (cardia versus others)
Body	3 (9%)	1 (5%)	0 (0%)	
Pyloric antrum	9 (28%)	4 (21%)	0 (0%)	
**MSI status**				
MSI-H	3 (9%)	1 (5%)	0 (0%)	1
MSS	29 (91%)	18 (95%)	4 (100%)	
**Pathological response**				
Complete	7 (22%)	4 (21%)	2 (50%)	0.18
Non-complete	25 (78%)	15 (79%)	2 (50%)	
**Relapse**				
Yes	8 (25%)	7 (37%)	1 (25%)	0.52
No	24 (75%)	12 (63%)	3 (75%)	

Tumor staging was performed according to the 7th UICC TNM classification [21] using endoscopic ultrasound, CT scan, PET/CT and laparoscopic staging. MSI: microsatellite instability; MSI-H: microsatellite instable; MSS: microsatellite stable; N.S.: not significant.

**Table 2 cancers-11-00396-t002:** List of trackable mutations, mutant allelic frequency (MAF) of trackable mutations in tissue and circulating tumor DNA according to time point.

P	Gene	Mutation		Tumor Analysis		Circulating Tumor DNA Detection by ddPCR		
				MAF NGS	MAF ddPCR	MAF Before CT	MAF After CT	MAF After Surgery
1	*TP53*	c.158G>A	p.W53X	26.5	26.0	0	0	NA
2	*TP53*	c.844C>T	p.R282W	13.4	20.0	0	0	0
3	*TP53*	c.637C>T	p.R213X	18.8	20.0	0	0	NA
4	*TP53*	c.817C>T	p.R273C	13.3	13.5	0	0	NA
5	*TP53*	c.743G>A	p.R248Q	11.5	16.0	**2.1**	0	0
6	*TP53*	c.743G>A	p.R248Q	13.7	18.2	0	0	**0.7**
7	*TP53*	c.844C>T	p.R282W	11.3	16.1	0	0	0
8	*PIK3CA*	c.3140A>G	p.H1047R	3.5	3.7	0	0	0
9	*TP53*	c.536A>G	p.H179R	31.6	30.0	0	0	0
10	*TP53*	c.844C>T	p.R282W	10.3	NA	**1.16**	0	NA
11	*TP53*	c.524G>A	p.R175H	11.1	11.5	0	0	NA
12	*KRAS*	c.38G>A	p.G13D	12.4	12.0	0	NA	0
13	*TP53*	c.810T>G	p.F270L	42.6	48.6	0	0	0
14	*TP53*	c.724T>C	p.C242R	15.5	16.1	0	0	0
15	*TP53*	c.451C>T	p.P151S	30.0	24.0	0	0	0
16	*TP53*	c.535C>T	p.H179Y	16.1	15.7	0	0	0
17	*TP53*	c.733G>A	p.G245S	5.6	6.2	**0.8**	0	0
18	*TP53*	c.659A>G	p.Y220C	6.9	5.6	**2.3**	0	0
19 *	*ATM*	c.5644C>T	p.R1882X	15.8	13.4	0	0	NA
19 *	*CTNNB1*	c.110C>T	p.S37F	5.3	4.0	0	0	NA

P: patient; NGS: next-generation sequencing; CT: chemotherapy; ddPCR: droplet digital PCR; NA: not available. * Two trackable mutations were tested for patient 19. Bold highlights positive value.

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
