# Peer review of "Limited Sensitivity of Circulating Tumor DNA Detection by Droplet Digital PCR in Non-Metastatic Operable Gastric Cancer Patients"

_cancers, 2019, doi:10.3390/cancers11030396_

Round 1

Reviewer 1 Report

The authors present a next-gen sequencing of circulating tumor DNA from gastric cancer patients.  The work is interesting, and they have done a great deal of good work.  The following could strengthen the manuscript.

The manuscript lacks many details on the experimental procedure, specifically DNA isolation.

The statistical analysis does not discuss uncertainty, especially figure 2.

Author Response

Reviewer 1

The authors present a next-gen sequencing of circulating tumor DNA from gastric cancer patients.  The work is interesting, and they have done a great deal of good work.  The following could strengthen the manuscript.

The manuscript lacks many details on the experimental procedure, specifically DNA isolation.

In this study we used QiaSymphony SP instrument and circulating DNA kit (QIAGEN), which is an automated technique.

We have specified in methods:” Cell-free circulating DNA (cfcDNA) extraction was performed on 4 mL of plasma using the QiaSymphony SP instrument and the QIAsymphony Circulating DNA kit (Qiagen), an automated technique, according to the manufacturer’s protocol”

The statistical analysis does not discuss uncertainty, especially figure 2

We have added CI95% in figure 2 CI95%

We have added in results CI95%:

At baseline, 4/19 patients had detectable ctDNA (>0.1%); the ddPCR technique displayed a sensitivity of 21% (95%CI=[8.5-43%], in this non-metastatic setting (Table 2)...

ctDNA was undetectable in all patients (0%, n=0/18, 95%CI= [0%-18%]), including those with detectable ctDNA at baseline (Figure 2). After surgery, ctDNA was detectable in one patient (n=1/13, 95%CI= [1%-33%]) with a MAF of 0.7%.

Reviewer 2 Report

The proof-of-principle study by Cabel et al. describes the efforts of the authors to examine ctDNA from patients with non-metastatic gastric cancer. Their pilot study included 32 patients. After NGS with a 39 gene tumor panel, the authors identified trackable mutations in 20 patients. Plasma was available in 19 patients before chemotherapy, in 18 before and 13 after surgery. While the patient numbers are low, this pilot study indicates that ctDNA, as studied by these authors, is not predictive of efficacy of chemotherapy and subsequent relapse.

This reviewer congratulates the authors for reporting ‘negative’ findings. In the important assessment of liquid biopsies, the study of ctDNA in cancer is important. If it cannot be used as a clinical tool in gastric cancer, such knowledge needs to be widely shared.

Prior to publication, however, a few important points should be considered.

1)    While the data presented are clear, the discussion should address the following potentially limiting points;

a)     was the selection of ‘trackable’ mutations an issue that created bias? The 39 gene tumor panel may have been too restrictive. What if whole exome sequencing was performed for each tumor sample? Would this have identified patient-specific makers that could have been traced in the plasma?

b)    Can the authors discuss the use of cell-free RNA, exosomes and/or CTCs in gastric cancer? What has been found to date and could these liquid biopsy types add to the monitoring of gastric cancer patients?

2)    The title of the manuscript is not specific enough. It should indicate that the monitoring of ctDNA is not seen as a monitoring tool for gastric cancer.

3)    The abstract does not convey the final message of the authors that they do not recommend ctDNA for monitoring of non-metastatic gastric cancer. This needs to be revised.

4)    Minor comment 1: Table S1 was not included in the manuscript download from the website.

5)    Minor comment 2: some text is highlighted in green. I do not know why.

Author Response

1)    While the data presented are clear, the discussion should address the following potentially limiting points;

a)     was the selection of ‘trackable’ mutations an issue that created bias? The 39 gene tumor panel may have been too restrictive. What if whole exome sequencing was performed for each tumor sample? Would this have identified patient-specific makers that could have been traced in the plasma?

We have added in the discussion:

More sensitive detection techniques, such as ultra-deep next-generation sequencing using unique molecular identifiers bar-coding strategy may need to be used[25]. However, the mechanisms underlying this low detection rate have yet to be determined. By using a broader panel of genes for sequencing, such as whole-exome sequencing, more patients could have benefited from ctDNA monitoring (12 patients in this study could not be monitored). In addition, several trackable mutations could have been detected for each patient, which could increase the detection sensitivity. It is nevertheless important to monitor mutations considered as drivers, because passenger mutations evolution is less representative of tumor evolution. Also, patients with different tumor mutations may have a different clinical course, as these mutations can be prognostic

b)    Can the authors discuss the use of cell-free RNA, exosomes and/or CTCs in gastric cancer? What has been found to date and could these liquid biopsy types add to the monitoring of gastric cancer patients?*

We have added in the discussion

“Another approach to ctDNA detection based on detection of circulating tumor methylation could be of interest in this context 27,28. In addition to ctDNA, other circulating tumor biomarkers, such as circulating tumor cells (CTC) and exosomes might be also relevant. The detection rate of CTC, reported in multiple studies29–32, appears however low in metastatic gastric cancer patients (e.g. less than half of 106 patients with advanced gastric cancer displayed ≥2 CTCs using CellSearch33), and precludes any clinical use in non-metastatic gastric cancer patients. Lastly, exosome detection has gained increasing interest, being potentially more sensitive than ctDNA in lung cancer34, but only few preliminary reports are available in gastric cancers35,36”

2)    The title of the manuscript is not specific enough. It should indicate that the monitoring of ctDNA is not seen as a monitoring tool for gastric cancer.

We have changed the title to:” Limited sensitivity of circulating tumor DNA detection by droplet digital PCR in non-metastatic operable gastric cancer patients”

3)    The abstract does not convey the final message of the authors that they do not recommend ctDNA for monitoring of non-metastatic gastric cancer. This needs to be revised.

We have changed the conclusion of the abstract to:” ctDNA monitoring during preoperative chemotherapy and after surgery does not appear to be a useful tool in clinical practice in non-metastatic gastric cancer to predict the efficacy of chemotherapy and subsequent relapse, essentially due to the poor sensitivity of ctDNA detection

4)    Minor comment 1: Table S1 was not included in the manuscript download from the website.

We have uploaded it

5)    Minor comment 2: some text is highlighted in green. I do not know why.

We have corrected it

Round 2

Reviewer 1 Report

The authors have addressed concerns about statistics.  However, simply stating that the DNA detection method is an automated technique is not sufficient.  Please detail the experimental protocols, as already requested.

Author Response

We apologize for our mistake and have detailed the ddPCR method for ctDNA detection

"Briefly ddPCR assays matching the trackable somatic mutations found in tumor tissue were purchased from Bio-Rad and the analysis were performed using the dedicated platform develop by bio-Rad according to the manufacturer’s protocol. Briefly, for each sample, a total volume of 20 μL PCR reaction mixture was prepared as 10 μL 2x Supermix for Probes without dUTP (Bio-Rad), 1 μL 20x target primers/probe (Bio-Rad), 1 μL 20x wild-type primers/probe (Bio-Rad) and DNA sample/water qsp 20 µL. The PCR reaction mixture was portioned into droplets using the QX100 Droplet Generator (Bio-Rad) according to the manufacturer’s instruction. The Droplets were then transferred to a 96-well PCR plate and the ddPCR was done using a C1000 Thermal Cycler (Bio-Rad) as follows: 95°C for 10 min, 40 cycles of 94°C for 30 sec, 55°C for 60 sec and 10 min at 98°C. The samples were then transferred to a Bio-Rad QX-100 droplet reader and analyzed on the basis of fluorescence intensity. The mutant allele frequency (MAF) was calculated for each sample using QuantaSoft v1.7.4 software (Bio-Rad). Each test contained at least one negative control well with no DNA and the corresponding tumor. ddPCR assays performances and detection threshold used in this study have been either reported elsewhere[17,25]  or validated by the manufacturer (Bio-Rad)"

Reviewer 2 Report

The authors have revised their manuscript, and it is now acceptable for publication.

Author Response

Thank you for your time